# Image Modeling with Deep Convolutional Gaussian Mixture Models

## Abstract

In this conceptual work, we present Deep Convolutional Gaussian Mixture Models (DCGMMs), a deep hierarchical Gaussian Mixture Model (GMM) that is particularly suited for describing and generating images. Vanilla (i.e., flat) GMMs require a very large number of components to well describe images, leading to long training times and memory issues. DCGMMs avoid this by a stacked architecture of multiple GMM layers, linked by convolution and pooling operations. This allows to exploit the compositionality of images in a similar way as deep CNNs do. DCG-MMs can be trained end-to-end by Stochastic Gradient Descent. This sets them apart from vanilla GMMs which are trained by Expectation-Maximization, requiring a prior k-means initialization which is infeasible in a layered structure. For generating sharp images with DCGMMs, we introduce a new gradient-based technique for sampling through non-invertible operations like convolution and pooling. Based on the MNIST and FashionMNIST datasets, we validate the DCGMM model by demonstrating its superiority over flat GMMs for clustering, sampling and outlier detection.

## 1 Introduction

This conceptual work is in the context of probabilistic image modeling, whose main objectives are density estimation and image generation (sampling). Since images usually do not precisely follow a Gaussian mixture distribution, such a treatment is inherently approximative in nature. This implies that clustering, even though it is possible and has a long history in the context of Gaussian Mixture Models (GMMs), is not a main objective. Sampling is an active research topic mainly relying on Generative Adverserial Networks (GANs) discussed in Section 1.2. Similar techniques are being investigated for generating videos (Ghazvinian Zanjani et al., 2018; Piergiovanni & Ryoo, 2019).

An issue with GANs is that their probabilistic interpretation remains unclear. This is outlined by the fact that there is no easy-to-compute probabilistic measure of the current fit-to-data that is optimized by GAN training. Recent evidence seems to indicate that GANs may not model the full image distribution as given by training data (Richardson & Weiss, 2018). Besides, images generated by GANs appear extremely realistic and diverse, and the GAN model has been adapted to perform a wide range of visually impressive functionalities.

In contrast, GMMs explicitly describe the distribution $p(\boldsymbol{X})$, given by a set of training data $\boldsymbol{X} = \{\boldsymbol{x}_n\}$, as a weighted mixture of $K$ Gaussian component densities $\mathcal{N}(\boldsymbol{x}; \boldsymbol{\mu}_k, \boldsymbol{\Sigma}_k) \equiv \mathcal{N}_k(\boldsymbol{x})$: $p(\boldsymbol{x}) = \sum_k^K \pi_k \mathcal{N}_k(\boldsymbol{x})$. GMMs require the mixture weights to be normalized: $\sum_k \pi_k = 1$ and the covariance matrices to be positive definite: $\boldsymbol{x}^T \boldsymbol{\Sigma}_k \boldsymbol{x} > 0 \ \forall \boldsymbol{x}$. The quality of the current fit-to-data is expressed by the log-likelihood

$$\mathcal{L}(\boldsymbol{X}) = \mathbb{E}_n \left[ \log \sum_k \pi_k \mathcal{N}_k(\boldsymbol{x}_n) \right], \tag{1}$$

which is what GMM training optimizes, usually by variants of Expectation-Maximization (EM) (Dempster et al., 1977). It can be shown that arbitrary distributions can, given enough components, be approximated by mixtures of Gaussians (Goodfellow et al., 2016). Thus, GMMs are guaranteed to model the complete data distribution, but only to the extent allowed by the number of components $K$.

In this respect, GMMs are similar to flat neural networks with a single hidden layer: although, by the universal approximation theorem of Pinkus (1999) and Hornik et al. (1989), they *can* approximate arbitrary functions (from certain rather broad function classes), they fail to do so in practice. The reason for this is that the number of required hidden layer elements is unknown, and usually beyond the reach of any reasonable computational capacity. For images, this problem was largely solved by introducing deep Convolutional Neural Networks (CNNs). CNNs model the statistical structure of images (hierarchical organization and translation invariance) by chaining multiple convolution and pooling layers. Thus the number of parameters without compromising accuracy can be reduced.

## 1.1 OBJECTIVE, CONTRIBUTION AND NOVELTY

The objectives of this article are to introduce a GMM architecture which exploits the same principles that led to the performance explosion of CNNs. In particular, the genuinely novel characteristics are:

- formulation of GMMs as a deep hierarchy, including convolution and pooling layers,
- end-to-end training by SGD from random initial conditions (no k-means initialization),
- generation of realistic samples by a new sharpening procedure,
- better empirical performance than vanilla GMMs for sampling, clustering and outlier detection.

In addition, we provide a publicly available TensorFlow implementation which supports a Keras-like flexible construction of Deep Convolutional Gaussian Mixture Models instances.

## 1.2 RELATED WORK

**Generative Adverserial Networks** The currently most widely used models of image modeling and generation are Generative Adverserial Networks (Arjovsky et al., 2017; Mirza & Osindero, 2014; Goodfellow et al., 2014). Variational Autoencoders (VAEs) follow the classic autoencoder principle (Kingma & Welling, 2013), trying to reconstruct their inputs through a bottleneck layer, whose activities are additionally constrained to have a Gaussian distribution. GANs are trained adversarially, mapping Gaussian noise to image instances, while trying to fool an additional discriminator network, which in turn aims to distinguish real from generated samples. They are capable of generating photo-realistic images (Richardson & Weiss, 2018), although their probabilistic interpretation remains unclear since they do not possess a differentiable loss function that is minimized by training. They may suffer from what is termed *mode collapse*, which is hard to detect automatically due to the absence of a loss function. Due to their ability to generate realistic images, they are prominently used in models of continual learning (Shin et al., 2018).

**Hierarchical GMMs** Mixture of Factor Analyzers (MFAs) models (McLachlan & Peel, 2005; Ghahramani & Hinton, 1997) can be considered as hierarchical GMMs because they are formulated in terms of a lower-dimensional latent-variable representation, which is mapped to a higher-dimensional space. The use of MFAs for describing natural images is discussed in detail in Richardson & Weiss (2018), showing that the MFA model alone, without further hierarchical structure, compares quite favorably to GANs when considering image generation. A straightforward hierarchical extension of GMMs is presented by Liu et al. (2002) with the goal of unsupervised clustering: responsibilities of one GMM are treated as inputs to a subsequent GMM, together with an adaptive mechanism that determines the depth of the hierarchy. Garcia et al. (2010) present a comparable, more information-theoretic approach. A hierarchy of MFA layers with sampling in mind is presented by Viroli & McLachlan (2019), where each layer is sampling values for the latent variables of the previous one, although transformations between layers are exclusively linear. Van Den Oord & Schrauwen (2014) and (Tang et al., 2012) pursue a similar approach. All described approaches use (quite complex) extensions of the EM algorithm initialized by k-means for training hierarchical GMMs, except Richardson & Weiss (2018) use Stochastic Gradient Descent (SGD), although with a k-means initialization. None of these models consider convolutional or max-pooling operations which have been proven to be important for modeling the statistical structure of images.

**Convolutional GMMs** The only work we could identify proposing to estimate hierarchical convolutional GMMs is Ghazvinian Zanjani et al. (2018), although the article described a hybrid model where a CNN and a GMM are combined.

**SGD and End-to-End GMM Training** Training GMMs by SGD is challenging due to local optima and the need to enforce model constraints, most notably the constraint of positive-definite covariance matrices. This has recently been discussed in Hosseini & Sra (2020), although the proposed solution requires parameter initialization by k-means and introduces several new hyper-parameters and is, thus, unlikely to work as-is in a hierarchical structure. An SGD approach that achieves robust convergence even without k-means-based parameter initialization is presented by Gepperth & Pflb (2020). Undesirable local optima caused by random parameter initialization are circumvented by an adaptive annealing strategy.

## 2  DATA

For the evaluation we use the following image data sets:

**MNIST** (LeCun et al., 1998) is the common benchmark for computer vision systems and classification problems. It consists of $60\,000$ $28 \times 28$ gray scale images of handwritten digits (0-9).
**FashionMNIST** (Xiao et al., 2017) consists of images of clothes in 10 categories and is structured like the MNIST dataset.

Although these datasets are not particularly challenging for classification, their dimensionality of $784$ is at least one magnitude higher than datasets used for validating other hierarchical GMM approaches in the literature.

## 3  DCGMM: MODEL OVERVIEW

The Deep Convolutional Gaussian Mixture Model is a hierarchical model consisting of *layers* in analogy to CNNs.[1] Each layer with index $L$ expects an input tensor $\mathbf{A}^{(L-1)} \in \mathbb{R}^4$ of dimensions $N, H^{(L-1)}, W^{(L-1)}, C^{(L-1)}$ and produces an output tensor $\mathbf{A}^{(L)} \in \mathbb{R}^4$ of dimensions $N, H^{(L)}, W^{(L)}, C^{(L)}$. Layers can have internal variables $\boldsymbol{\theta}^{(L)}$ that are adapted during SGD training.

An DCGMM layer $L$ has two basic operating modes (see Figure 1): for (density) *estimation*, an input tensor $\mathbf{A}^{(L-1)}$ from layer $L-1$ is transformed into an output tensor $\mathbf{A}^{(L)}$. For *sampling*, the direction is reversed: each layer receives a control signal $\mathbf{T}^{(L+1)}$ from layer $L+1$ (same dimensions as $\mathbf{A}^{(L)}$), which is transformed into a control signal $\mathbf{T}^{(L)}$ to layer $L$-1 (same dimensions as $\mathbf{A}^{(L-1)}$).

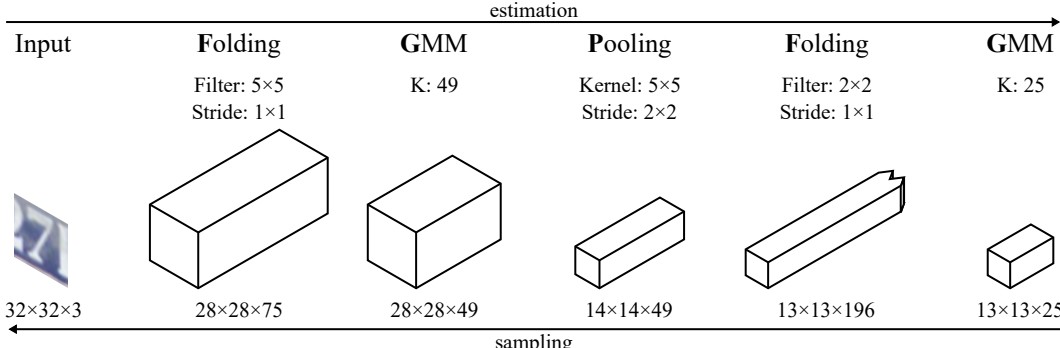

Figure 1: Illustration of a sample DCGMM instance containing all four layer types, with exemplary dimensionalities and parameters for each layer.

### 3.1  LAYER TYPES

We define three layer types: Folding (F), Pooling (P) and convolutional GMM (G). Each layer implements distinct operations for each of the two modes, i.e., estimation and sampling.

---

[1]TensorFlow code is available under `https://github.com/iclr2021-dcgmm/dcgmm`

**Folding Layer** For density estimation, this layer performs a part of the well-known convolution operation known from CNNs. Based on the filter sizes $f_X^{(L)}$, $f_Y^{(L)}$ as well as the filter strides $\Delta_X^{(L)}$, $\Delta_Y^{(L)}$, all entries of the input tensor inside the range of the sliding filter window are dumped into the channel dimension of the output tensor. We thus obtain an output tensor of dimensions $N, H^{(L)} = 1 + \frac{H^{(L-1)} - f_Y^{(L)}}{\Delta_Y^{(L)}}$, $W^{(L)} = 1 + \frac{W^{(L-1)} - f_X^{(L)}}{\Delta_X^{(L)}}$ and $C^{(L)} = C^{(L)} f_X^{(L)} f_Y^{(L)}$, whose entries are computed as $A_{nhwc}^{(L)} = A_{nh'w'c'}^{(L-1)}$ with $h = h'/f_Y^{(L)}$, $w = w'/f_X^{(L)}$ and $c = c' + \left( (h' - h\Delta_Y^{(L)}) f_X^{(L)} + w' - w\Delta_X^{(L)} \right) C^{(L-1)} + c'$. When sampling, it performs the inverse mapping which is not a one-to-one correspondence: input tensor elements which receive several contributions are simply averaged over all contributions.

**Pooling Layer** For density estimation, pooling layers perform the same operations as standard (max-)pooling layers in CNNs based on the kernel sizes $k_Y^{(L)}$, $k_X^{(L)}$ and strides $\Delta_X^{(L)}$, $\Delta_Y^{(L)}$. When sampling, pooling layers perform a simple nearest-neighbor up-sampling by a factor indicated by the kernel sizes and strides.

**GMM Layer** This layer type contains $K$ GMM components, each of which is associated with trainable parameters $\pi_k$, $\boldsymbol{\mu}_k$ and $\boldsymbol{\Sigma}_k$, $k = 1 \ldots K$, representing the GMM *weights*, *centroids* and *covariances*. What makes GMM layers convolutional is that they do not model single input vectors, but the channel content at *all* positions $h, w$ of the input $\mathbf{A}_{n,w,h,:}^{(L-1)}$, using a shared set of parameters. This is analog to the way a CNN layer models image content at all sliding window positions using the same filters. A GMM layer thus maps the input tensor $\mathbf{A}^{(L-1)} \in \mathbb{R}^{N, H^{(L-1)}, W^{(L-1)}, C^{(L-1)}}$ to $\mathbf{A}^{(L)} \in \mathbb{R}^{N, H^{(L-1)}, W^{(L-1)}, K}$, each GMM component $k \in \{0, \ldots, K\}$ contributing the likelihood $A_{nhwk}^{(L)}$ of having generated the channel content at position $h, w$ (for sample $n$ in the mini-batch). This likelihood is often referred to as *responsibility* and is computed as

$$p_{nhwk}(\mathbf{A}^{(L-1)}) = \mathcal{N}_k(\mathbf{A}_{nhw:}^{(L-1)}; \boldsymbol{\mu}_k, \boldsymbol{\Sigma}_k)$$
$$A_{nhwk}^{(L)} \equiv \frac{p_{nhwk}}{\sum_{c'} p_{nhwc'}}. \tag{2}$$

For training the GMM layer, we optimize the GMM log-likelihood $\mathcal{L}^{(L)}$ for each layer $L$:

$$\mathcal{L}_{hw}^{(L)} = \sum_n \log \sum_k \pi_k p_{nhwk}(\mathbf{A}^{(L-1)})$$
$$\mathcal{L}^{(L)} = \frac{\sum_{hw} \mathcal{L}_{hw}^{(L)}}{H^{(L-1)} W^{(L-1)}} \tag{3}$$

Training is performed by SGD according to the technique, and with the recommended parameters, presented by Gepperth & Pflb (2020), which uses a max-component approximation to $\mathcal{L}^{(L)}$. In sampling mode, a control signal $\mathbf{T}^{(L)}$ is produced by standard GMM sampling, performed separately for all positions $h, w$. GMM sampling at position $h, w$ first selects a component by drawing from a multinomial distribution. If the GMM layer is the last layer of a DCGMM instance, the multinomial's parameters are the mixing weights $\pi_:$ for each position $h, w$. Otherwise, the control signal $\mathbf{T}_{nhw:}^{(L+1)}$ received from layer $L + 1$ is used. It is consistent to use the control signal for component selection in layer $L$, since it was sampled by layer $L + 1$, which was in turn trained on the component responsibilities of layer $L$. The selected component (still at position $h, w$) then samples $\mathbf{T}_{nhw:}^{(L)}$. It is often beneficial for sampling to restrict component selection to the $S$ components with the highest control signal (top-$S$ sampling).

## 3.2 Architecture-Level Functionalities

The DCGMM approach proposes several functionalities on different architectural levels.

### 3.2.1 END-TO-END TRAINING

To train an DCGMM instance, we optimize $\mathcal{L}^{(L)}$ for each GMM layer $L$ by vanilla SGD[2], using learning rates $\epsilon^{(L)}$. This is different from a standard CNN classifier, where only a single loss function is minimized, usually a cross-entropy loss computed from the last layer's outputs. Learning is *not* conducted layer-wise but end-to-end. Parameter initialization for GMM layers selects the initial values for the mixing weights as $\pi_k = K^{-1}$, centroid elements sampled from $\boldsymbol{\mu}_{kl} \sim \mathcal{U}_{[-0.01,0.01]}$ and diagonal covariances are initialized to unit entries. To ensure good convergence, training is conducted in two phases, in the first one of which only centroids are adapted.

### 3.2.2 DENSITY ESTIMATION AND HIERARCHICAL OUTLIER DETECTION

Outlier detection requires the computation of long-term averages in all layers and positions, $\mathbb{E}_n\mathcal{L}^{(L)}_{nhw}$ and variances $\mathrm{Var}_n(\mathcal{L}^{(L)}_{nhw})$ over the training set, preferably during a later, stable part of training. Thus, for every layer and position $h, w$, inliers are characterized by

$$\mathcal{L}^{(L)}_{hw} \geq B^{(L)}_{hw} \equiv \mathbb{E}_n\mathcal{L}^{(L)}_{nhw} - c\sqrt{\mathrm{Var}_n(\mathcal{L}^{(L)}_{nhw})}. \tag{4}$$

Larger values of $c$ imply a less restrictive characterization of inliers.

Assuming that the topmost GMM layer is global ($h = w = 1$), Equation 4 reduces to a single condition that determines whether the sample, as a whole, is an inlier. However, we can also localize inlier/outlier image parts by evaluating Equation 4 in lower GMM layers.

### 3.2.3 SAMPLING AND SHARPENING

Sampling starts in the highest layer, assumed to be an GMM layer, and propagates control signals downwards (see Figure 1 and Section 3.1), with control signal $\mathbf{T}^{(1)}$ constituting the sampling result. Sampling suffers from information loss due to the not-invertible mappings effected by Pooling and Folding layers. To counteract this, each Folding layer $l$ performs *sharpening* on the generated control signal $\mathbf{T}^{(l)}$. This involves computing $\mathcal{L}^{(L^*)}(\mathbf{T}^{(L)})$ of the next-highest GMM layer at level $L^* > l$, and performing $G$ gradient ascent steps $T^{(L)}_{nhwc} \rightarrow T^{(L)}_{nhwc} + \epsilon_s \partial \mathcal{L}^{(L^*)}/\partial T^{(L)}_{nhwc}$. The reason for sharpening is that filters in Folding layers usually overlap, and neighboring filter results are correlated. This correlation is captured by all higher GMM layers, and most prominently by the next-highest one. Therefore, modifying $\mathbf{T}^{(L)}$ by gradient ascent will recover some of the information lost by pooling and folding. After sharpening, the tensor $\mathbf{T}^{(L)}$ is passed as signal to $L - 1$.

## 4 EXPERIMENTS

We define various DCGMM instances (with 2 or 3 GMM layers) for evaluation, see Table 1, plus a single-layer DCGMM baseline which is nothing but a vanilla GMM. A DCGMM instance is defined by the parameters of its layers: **F**olding($f_Y$, $f_X$, $\Delta_Y$, $\Delta_X$), (Max-)**P**ooling($k_Y$, $k_X$, $\Delta_Y$, $\Delta_X$) and **GMM**($K$). Unless stated otherwise, training is always conducted for 25 epochs, using the recommended parameters from (Gepperth & Pflb, 2020). Sharpening is always performed for $G = 1\,000$ iterations with a step size of $0.1$.

### 4.1 SAMPLING, SPARSITY AND INTERPRETABILITY

We show that trained DCGMM parameters are sparse and have a intuitive interpretation in terms of sampling. To this effect, we train DCGMM instance $2L$-$a$ (see Table 1). After training (see Section 3.2.1), we plot and interpret the centroids of the GMM layers 2 (G2) and 4 (G4). The centroids of layer 2 (left of Figure 2) are easily interpretable and reflect the patterns that can occur in any of the $2 \times 2$ input patches to layer G2 of size $20 \times 20$. The $36 = 6 \times 6$ centroids of G4 (right of Figure 2) express typical responsibility patterns computed from each of the $2 \times 2$ input patches to G2, and can be observed to be very sparsely populated. Another interpretation of G4

---

[2]Advanced SGD strategies like RMSProp (Hinton et al., 2012) or Adam (Kingma & Ba, 2015) seem incompatible with GMM optimization.

Table 1: Configurations and parameters of different DCGMM architectures.

| ID / layer | 1L | 2L-a | 2L-b | 2L-c | 2L-d | 2L-e | 3L-a | 3L-b |
|---|---|---|---|---|---|---|---|---|
| 1 | F(28,28,1,1) | F(20,20,8,8) | F(7,7,7,7) | F(7,7,3,3) | F(28,28,1,1) | F(7,7,3,3) | F(3,3,1,1) | F(28,28,1,1) |
| 2 | G(25) | G(25) | G(25) | G(25) | G(25) | G(25) | G(25) | G(25) |
| 3 |  | F(2,2,1,1) | F(4,4,1,1) | F(8,8,1,1) | F(1,1,1,1) | F(8,8,1,1) | P(2,2) | F(1,1,1,1) |
| 4 |  | G(36) | G(36) | G(36) | G(36) | G(36) | F(4,4,1,1) | G(25) |
| 5 |  |  |  |  |  |  | G(25) | F(1,1,1,1) |
| 6 |  |  |  |  |  |  | P(2,2) | G(25) |
| 7 |  |  |  |  |  |  | F(6,6,1,1) |  |
| 8 |  |  |  |  |  |  | G(49) |  |
| *comment* | vanilla GMM | 1 conv. layer | 1 conv. layer | 1 conv. layer | no convolutions | 1 conv. layer | 2 conv. layers | no convolutions |

centroids can be found in terms of sampling (see Section 3.2.3), which would first select a random G4 component to generate a sample of dimensions $H, W, C = 1, 1, 2 \times 2 \times 5 \times 5$ from it, and pass it on as a control signal to G2. Traversing Folding layer 3 only reshapes the control signal to dimension $H, W, C = 2, 2, 5 \times 5$, depicted in the middle of Figure 2. This signal controls component selection in each of the $2 \times 2$ positions in G2: due to their sparsity, we can directly read off the components likely to be selected for sampling at each position. G2 thus generates a control signal whose $2 \times 2$ positions of dimensions $H, W, C = 20, 20, 1$ overlap in the input plane (this is resolved by sharpening in Folding layer 1). In this case, it is easy to see that sampling produces a particular representation of the digit zero.

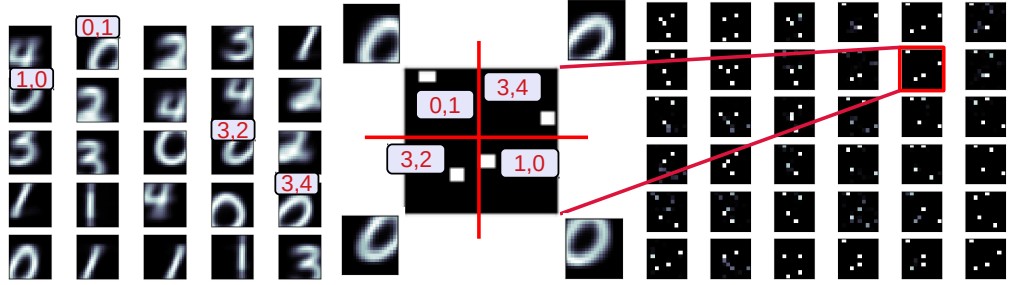

Figure 2: Sampling from DCGMM instance $2L$-$a$, see Table 1. Shown are learned GMM centroids (left: G2, right: G4, see text) and an illustration of sampling, having initially selected the layer 4 component highlighted in red. In the middle, the selected G4 centroid is shown in more detail.

## 4.2 OUTLIER DETECTION

For outlier detection, we compare DCGMM architectures from Table 1, using the log-likelihood of the highest layer as a criterion as detailed in Section 3.2.2. We first train a DCGMM instance on classes 0-4, and subsequently use the trained classes for inlier- and class 5-9 for outlier-detection. We vary $c$ in the range $[-2, 2]$, resulting in different outlier and inlier percentages. Figure 3 shows the ROC-like curves thus clearly indicate that the deep convolutional DCGMM instances perform best, whereas deep but non-convolutional instances like $2L$-$d$ and $3L$-$b$ consistently perform badly.

## 4.3 CLUSTERING

We compare DCGMMs to vanilla GMMs using established clustering metrics, namely the Dunn index (Dunn, 1973) and the Davies-Bouldin score (Davies & Bouldin, 1979). The DCGMM instances from Table 1 are tested on both image datasets. We observe that mainly the deep but non-convolutional DCGMM instances perform well in clustering, whereas convolutional instances, even if they are deep, are compromised. Please note that these metrics do not measure the classification accuracy obtained by clustering but intrinsic clustering-related properties.

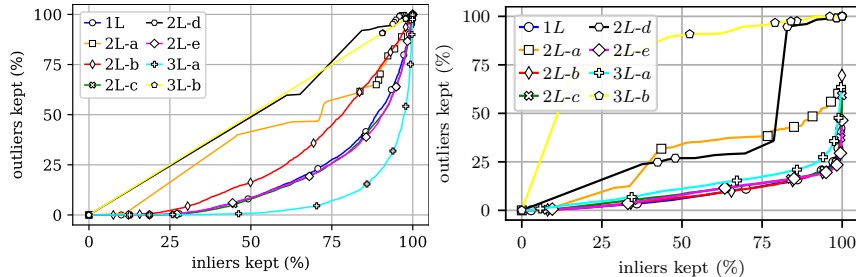

Figure 3: Visualization of different DCGMM architectures and its outlier detection capabilities for MNIST (left) and FashionMNIST (right).

Table 2: Two metrics, Dunn index (higher is better) and Davies-Bouldin (DB) score (smaller is better), evaluated for all tested DCGMM architectures on MNIST and FashionMNIST. Best results are marked in bold. The given numbers are worst cases over 10 independent runs.

| Dataset | DCGMM Metric | 1L | 2L-a | 2L-b | 2L-c | 2L-d | 2L-e | 3L-a | 3L-b |
|---------|--------------|-----|------|------|------|------|------|------|------|
| MNIST | Dunn index | 0.14 | 0.14 | 0.13 | 0.12 | **0.19** | | 0.15 | 0.15 |
| | DB score | 2.59 | 2.73 | 3.06 | 2.62 | 2.57 | | 2.65 | **2.53** |
| Fashion-MNIST | Dunn index | 0.14 | 0.15 | **0.16** | 0.15 | 0.11 | 0.11 | 0.096 | 0.13 |
| | DB score | 2.37 | 2.77 | 2.62 | 2.7 | 2.40 | 2.92 | 3.2 | **2.35** |

## 4.4 SAMPLING AND SHARPENING

The results presented here were obtained by training on classes 0-4 of both datasets, and have to be confirmed by visual inspection of generated samples.

**Effects of Convolution on Sampling Diversity** To assess this, we compare the samples generated by the non-convolutional DCGMM instances $1L$, $2L$-$d$ to the samples generated by the convolutional instances $2L$-$d$, $2L$-$e$ for top-1-sampling. The results of Figure 4 clearly indicate that convolutional DCGMMs generate diverse samples, whereas non-convolutional ones basically replicate their centroids with added white noise. FashionMNIST results are given in Appendix A.2.

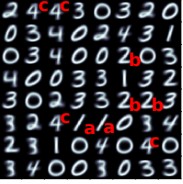 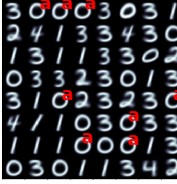 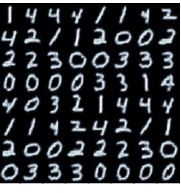 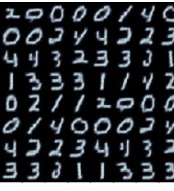

Figure 4: Sampling diversity: top-1 sampling shown on MNIST, from left to right, for DCGMM architectures $1L$ (vanilla GMM), $2L$-$d$ (non-convolutional 2-layer), $2L$-$c$ and $2L$-$e$ (convolutional 2-layer). Please observe duplicated samples in the non-convolutional architectures, marked in red.

**Generating Sharp Images** Figure 5 shows the effect of sharpening for DCGMM instance $2L$-$c$ using top-1-sampling. We can observe that the overall shape of a sample is not changed but that the outlines are crisper, an effect visible especially for FashionMNIST. Thus, sharpening does no harm and rather improves the visual quality of generated samples. See Appendix A.3 for FashionMNIST results.

**Controlling Diversity by Top-S-Sampling** Using instance $2L$-$c$, Figure 5 demonstrates how sample diversity is related to $S$: a higher value yields more diverse samples, but increases the risk of generating corrupted samples or outliers. As the corresponding FashionMNIST results in Appendix A.1 show, a good value of $S$ is clearly problem-dependent.

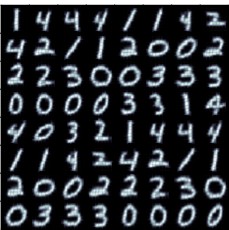 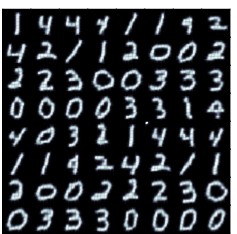 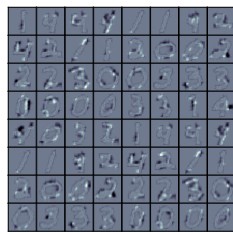

Figure 5: Impact of sharpening on top-S-sampling with $S$=1, see Section 3.2.3, shown for DCGMM instance $2L$-$c$ on MNIST. Shown are unsharpened samples (left), sharpened samples (middle) and differences (right). Samples at the same position were generated by the same top-level prototype.

# 5 SUMMARY, DISCUSSION AND CONCLUSION

The **Objective of the article** was to establish the conceptual foundations of deep GMM hierarchies (see also Appendix A) that leverage important mechanisms from the CNN domain. Convolution and pooling layers make it possible to apply the model to high-dimensional image data with off-the-shelf hardware (typical training runs take approximately 2 minutes on a GeForce GTX 1080).

**Important results** show the illustration of important functionalities such as outlier detection, clustering and sampling, which no other work on hierarchical GMMs can present related to such high-dimensional image datasets. We also propose a method to generated sharp images with GMMs, which has been a problem in the past (Richardson & Weiss, 2018). An interesting facet of our experimental results is that non-convolutional DCGMMs seem to perform better at clustering, whereas convolutional ones are better at outlier detection and sampling.

A **Key point** of the article is the compositionality in natural images. This property is at the root of DCGMM's ability to produce realistic samples with relatively few parameters. When considering top-S-sampling in a layer $L$ with $H^{(L)}W^{(L)} = P^{(L)}$ positions, the number of distinct control signals generated layer $L$ is $S^{P^{(L)}}$. A DCGMM instance with multiple GMM layers $\{L_i\}$ can thus sample $\prod_L S^{P^{(L)}}$ different patterns, which grows with the depth of the hierarchy *and* the number of distinct positions in a layer, making a strong argument in favor of deep convolutional hierarchies such as DCGMM. This is an argument similar to the one about different paths through a hierarchical MFA model in (Viroli & McLachlan, 2019), although their number grows more strongly for DCGMM.

**Differences to other hierarchical models** such as (Viroli & McLachlan, 2019; Van Den Oord & Schrauwen, 2014; Tang et al., 2012) are most notably the introduction of convolution and pooling layers. Our experimental validation can therefore be performed on high-dimensional data, such as images, with moderate computational cost, instead of low-dimensional problems such as the artificial Smiley task or the *Ecoli* and related problems. Our experimental validation does not exclusively focus on clustering performance (problematic with images) but on demonstrating the capacity for realistic sampling and outlier detection. Lastly, training DCGMMs by SGD facilitates efficient parallelizable implementations, as demonstrated with the provided TensorFlow implementation.

**Next steps** will consist of exploring the layered DCGMM architecture, mainly top-S-sampling, for generating natural images.

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

## A PROBABILISTIC INTERPRETATION OF DCGMMS

A probabilistic interpretation of the DCGMM model is possible despite its complex structure. The simple reason is that DCGMM instances produce outputs which are inherently normalizable, meaning that the integral over an infinite domain (e.g., data space) remains finite. Thus, DCGMM outputs can be interpreted as a probability which is not the case for DNN/CNNs due to the use of scalar products.

Here, we prove that GMMs are normalizable in the sense that the integral of the log-probability $\mathcal{L}(\boldsymbol{x}) = \log \sum_k \pi_k p_k(\boldsymbol{x})$ is finite. This holds for any GMM layer in a hierarchy regardless of its input, provided that the input is finite (which is assured because Pooling and Folding layers cannot introduce infinities). For simplicity, we integrate over the whole $d$-dimensional space $\mathbb{R}^d$. Since the component probabilities are Gaussian and thus strictly positive, and since furthermore the mixing weights are normalized and $\geq 0$, the sum is strictly positive. Thus, it is sufficient to show that the integral over the inner sum (the argument of the logarithm) is finite. We thus have

$$\int_{\mathbb{R}^d} \sum_k \pi_k p_k(\boldsymbol{x}) d\boldsymbol{x} = \sum_k \pi_k \int_{\mathbb{R}^d} p_k(\boldsymbol{x}) = \sum_k \pi_k \sqrt{\det(2\pi\Sigma)} \tag{5}$$

which is trivially finite because Gaussians are normalized.

### A.1 TOP-S SAMPLING RESULTS FOR FASHIONMNIST

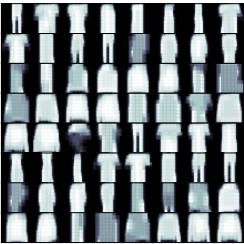 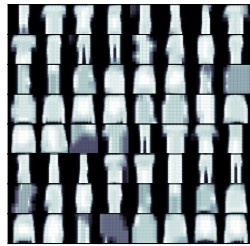 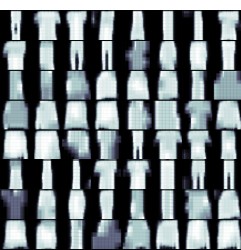

Figure 6: Impact of higher values of $S$ in top-S sampling, shown for DCGMM instance $2L$-$c$. From left to right: $S$=2,5,10.

### A.2 EFFECTS OF CONVOLUTION ON SAMPLING DIVERSITY FOR FASHIONMNIST

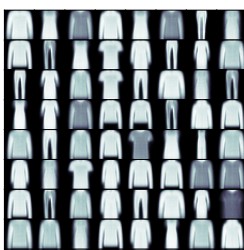 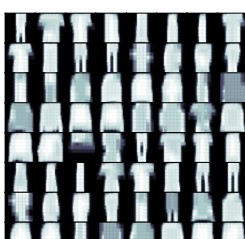 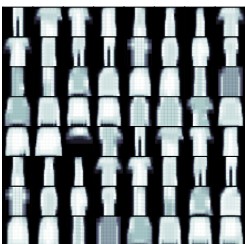

Figure 7: Convolutional architecture is helpful for sampling: top-1 sampling shown on Fashion-MNIST, from left to right, for DCGMM architectures $1L$ (vanilla GMM), $2L$-$d$ (non-convolutional 2-layer DCGMM), $2L$-$c$ and $2L$-$e$ (both convolutional 2-layer DCGMMs). Please observe duplicated samples in the non-convolutional architectures.

## A.3 EFFECTS OF SHARPENING ON FASHIONMNIST

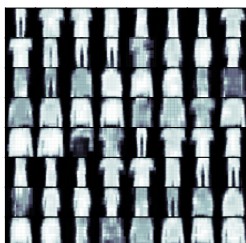 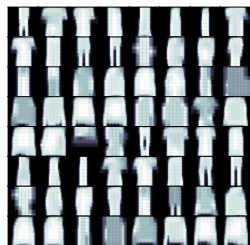 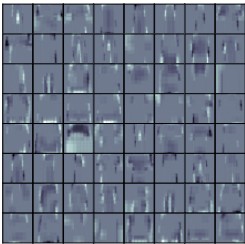

Figure 8: Impact of sharpening on top-S-sampling with $S$=1, see Section 3.2.3, shown for DCGMM instance $2L$-$c$ on FashionMNIST. Shown are unsharpened samples (left), sharpened samples (middle) and differences (right). Samples at the same position were generated by the same top-level prototype and are thus directly comparable.

