# OpenReview forum: "Image Modeling with Deep Convolutional Gaussian Mixture Models"
_ICLR.cc/2021/Conference — Reject_

### Official Review · AnonReviewer4 · 2020-10-21
**Heuristics shouldn't be considered a probabilistic model**

**Rating:** 2
**Confidence:** 5

**Review:**

Summary
-------------
This paper defines encoding and decoding procedures which use transformations inspired by Gaussian mixture models (GMMs). The decoding procedure further involves "sharpening" steps. A heuristic for training the parameters shared by the encoder and decoder is proposed which optimizes the likelihoods of GMMs defined on various outputs of the encoder. The decoder is evaluated in terms of clustering performance, sample quality, and outlier detection.

Quality (1/5)
-----------------
It seems like a stretch to call the proposed model a "deep Gaussian mixture model" or a probabilistic model at all. For a probabilistic model we should be able to assign a probability to any (measurable) set of the input space. (For some models such as GANs this probability is intractable but it is still easily defined.) However, it is not clear from the model's description (Section 3) what this probability should be. This lack of a well defined density (or measure) is surprising given the authors' emphasis on "density estimation" as a "main objective" of probabilistic image modeling and a potential application of their model.

One could view the entire decoding process as a complicated generative model which involves an iterative sharpening procedure, but this is not how the model is presented. In particular, the training procedure does not seem to be optimizing any divergence of this model and it is not clear how the encoder relates to the posterior of this model.

The output of the GMM layer ("responsibilities") live on a simplex (Eq. 2). If we stack two GMM layers, doesn't the likelihood of the second GMM explode (since the differential entropy of the inputs is negative infinity)?  Is suspect the reason that the training loss doesn't explode may be an artefact of SGD and/or the pooling layers.

Clarity (3/5)
----------------
I appreciated that the encoding and decoding procedure as well as the training objective were clearly described.

On the other hand, already in the abstract and the first two paragraphs the authors make confusing claims such as the following:

(1) The authors claim in the abstract that "DCGMMs can be trained end-to-end by SGD" and that this "sets them apart from vanilla GMMs which are trained by EM, requiring a prior k-means initialization". But vanilla GMMs may very well be trained with SGD as the authors note themselves in the related work section. While k-means may speed up training, it is not "required" by GMMs.

(2) The authors claim that since "images usually do not precisely follow a GMM distribution [...] clustering is not a main objective [of image modeling]." This seems like a non-sequitur.

(3) "An issue with GANs is that their probabilistic interpretation remains unclear. This is outlined by the fact that there is no easy-to-compute probabilistic measure of the current fit-to-data that is optimized by GAN training." I would argue that the divergence(s) (approximately) optimized by GANs as well as their probabilistic interpretation are much better understood than the proposed model, as discussed above.

The authors point out that "training GMMs by SGD is challenging" because the covariance matrices are constraint to be positive definite. Isn't reparametrization relatively easy (C = AA')? And don't you have the same issue in your model (Eq. 2)? How do you enforce positive definiteness in your model?

Originality (2/5)
---------------------
I would have expected a more thorough comparison with deep GMMs (van den Oord & Schrauwen, 2014) which appears to be the most closely related model.

Another line of research not being discussed are autoregressive Gaussian mixture models (e.g., Domke, 2008; Hosseini et al., 2011; Theis et al., 2012). These models generalize Gaussian mixture models and are able to efficiently model images of arbitrary size by (like convolutions) making a stationarity assumption. Deep extensions exist as well (Theis & Bethge, 2015).

Significance (1/5)
-----------------------
A lack of conceptual insights or a principled motivation would be fine if the empirical results made up for it. Unfortunately the empirical evaluation seems rather limited as well. No comparisons were made to previously published baselines. Instead, all results are only compared to other DCGMM results provided by the authors.

The chosen tasks and datasets (MNIST, FashionMNIST) are rather limited as well. In particular, unconditional image generation is not a well defined task (and certainly not a "main objective" of image modeling) but rather a (poor) proxy for evaluating generative models (see Theis et al., 2016).

---

### Official Review · AnonReviewer2 · 2020-10-26
**Needs comparison to other models and probabilistic interpretation seems unsure**

**Rating:** 3
**Confidence:** 4

**Review:**

In this manuscript the authors present a variant of stacked Gaussian mixture models they propose for modeling images called Deep Convolutional Gaussian Mixture Model. This model may contain analogues of convolutional layers and nonlinearities between the stacked Gaussian mixture models. This model can then be trained using stochastic gradient decent on the gradients propagated through the model. Finally the authors show some experimental evaluation on FashionMNIST and MNIST.

Overall I vote for rejection. While the model the authors present seems to work in principle on images I do not think the authors present a good argument why their model should be used for modeling images and there are definitely other models the authors should compare their model to. Also I have doubts whether the model as presented is a proper probabilistic model.

Pros:
1) This is a new Gaussian mixture based model.
2) It is stackable and inherits some of the benefits of DNNs.
3) It is trainable with (stochastic) gradient descent

Cons:
1) I disagree with the authors in the introduction. While GANs as they discuss are not fully probabilistic models and thus have limitations to their applicability, other models do have very clear probabilistic interpretations and apply to all the tasks discussed here. Examples of such networks are the numerous variations of the variational autoencoder, the invertable network based variations like FLOW or GLOW and diverse others. As these are ignored by the authors, I don’t think they place their work well into the literature and do not see a particularly strong argument here that gaussian mixture models would be a great addition to the modeling of images.

2) The authors do not compare their method against any competing methods outside the Gaussian mixture model framework. I think they would have to present some comparisons to state of the art methods for the tasks they test their model on. At very least compare to some basic models which generate some intuition where Gaussian mixture models overall lie in terms of performance. Without that I am completely lost whether the performance in these tasks is any good.

3) If the authors instead want to focus on the conceptual level or advancing our understanding of the presented kind of model I still think a lot more could and should be done: For example, the authors decide to perform outlier detection based solely on the last GMM layer. While this might be somewhat sensible for the decision between different numbers or categories in MNIST, in general, this is not the probability of the observed data under the model. Why is this used? Similarly: What is the structure of the representations produced by the model?

4) Given that the main claimed advantage of the model is its probabilistic interpretation, I find the probabilistic description and analysis of the model somewhat lacking:
    - How exactly is the probability of a given sample computed in this model?
    - How can convolution and pooling layers be interpreted as probabilistically given that they are not invertible. It seems to me, that for both types of layers, the input may even have zero probability to be produced by the described sampling processes. I.e. either some inputs have 0 probability under the model or the sampling methods do not actually sample from the model.
    - just ignoring pooling and convolution steps in the calculation of the probabilities under the model as I guess the authors do here seems wrong.

5) Each Gaussian mixture model layer in the proposed model converts a continuous input space into  a probability distribution over Group assignments. These assignment probabilities are then linearly mapped and pooled before another Gaussian mixture model layer again interprets the input as point in a R^n to be modeled by a mixture of Gaussians. This is technically possible to some degree if we ignore the restrictions on the support to achieve a valid distribution, but I do not get the intuition how this may well represent the composition of an image. It is stated as fact that the presented model is good for that, but I think this part requires some justification. Wouldn’t we expect operations which further work on distributions over discrete spaces instead?

---

### Official Review · AnonReviewer3 · 2020-10-28
**An interesting way to stack GMMs as density estimators  in a convolutional architecture**

**Rating:** 4
**Confidence:** 4

**Review:**

## Summary
The authors propose a convolutional neural network architecture where some layers realize a Gaussian mixture model (GMM) that performs density estimation over the embeddings of the previous layers it receives as inputs.
These embeddings collect probability values.
As such, their whole architecture, DCGMM, can be viewed as stacking density estimators, each of which is fit by maximum likelihood over the output embeddings of the preceding ones, i.e. it estimates a density on a series of latent spaces.
As such, when applied to images, DCGMM does not provide an explicit likelihood over the pixel space -- or better only its first layer, comprising a shallow GMM, does.
The authors use DCGMM as a simulator, i.e., to sample images and devise a sharpening scheme that tries to work around the non-invertibility of pooling layers (but also of the GMM layer?)

## Presentation
The paper is overall well-written and readable.
The biggest concern I have regarding presentation is that the proposed DCGMM is not a density estimator per se nor it is used for desity estimation in the experiments, while it is presented as such.
I would suggest authors a rewriting that includes a deep discussion that also properly contrasts DCGMM with other hierarchical models listed in Section 1.2. It seems to me that the latter define a proper density $p(X)$ over the observables $X$ while DCGMM leaves the task to the shallow GMM of its first layer.
In particular, the way to evaluate (and then train) the model, can be named 'forward evaluation' mode more than density estimation.

## Contributions
The main contribution, i.e., to perform stacked density estimation, seems novel to me.
However, I would advice authors to directly present it in this way and not as a normal density estimator over oversables.
This implies properly contrasting DCGMM w.r.t. the existing literature (see also comments above).
Deep (Gaussian) mixture models have been research in the literature [1,2,3]
DIfferently from DCGMM, they retain an explicit and tractable likelihoods (but also marginalization and conditioning) as well as the ability to exactly sampling from them.
These deep mixtures also show how it is possible to train a discrete latent variable model with thousands of latent variables effectively with EM [4]

I feel that the proposed sampling scheme needs more space for discussion and clarifications.
First, it is not clear to me if the proposed sampling scheme is consistently and correctly sampling from the space $X, Z_1, Z_2, ..., Z_D$ where the latter are the $D$ latent variable spaces associated to the GMM layers in DCGMM.
Second, I wonder if the GMM layer is invertible per se: two different incoming inputs can get associated the same likelihood by a single Gaussian due to the symmetry of its density. The same might happen when the GMM retains some symmetries as well.

Concerning the usage of DCGMM as an outlier detector, it is not clear why only the likelihoods on the top-most latent space ($Z_D$) are employed. I wonder what happens when all the layers are utilized (singularly or collectively).


## Experiments
The experiments focus on using DCGMM as a simulator and as an outlier detector (or clustering) for MNIST and FashionMNIST.
These count as interestnig preliminary results but go against the original motivation that DCGMM overcomes the limitations of other deep mixture models that are not able to scale to larger datasets.
Furthermore, it would be insightful to compare DCGMM -- even on MNIST and FashionMNIST alone -- against the deep mixture models referenced in Section 1.2.

For the sampling experiment, the effect of sharpening and/ top-S sampling could be measured in a more rigorous way.
For instance, the quality of samples can be assesed via FID scores or any other analogous metric from the GAN/VAE literature.
Moreover,  are 'duplicates' -- possibly signaling mode collapse -- exact replicas or slight variations? This can be further inspected by reporting for each generated sample its top-k nearest neighbor samples in the training set and some metric (even in pixel-space) of divergence between them.

Why are authors training only on classes "0-4"? Is there an issue in scalability?


## References

[1] Sharir, Or, et al. "Tensorial mixture models." AISTATS (2018).
[2] Jaini, Priyank, Pascal Poupart, and Yaoliang Yu. "Deep homogeneous mixture models: representation, separation, and approximation." Advances in Neural Information Processing Systems. 2018.
[3] Butz, Cory J., et al. "Deep convolutional sum-product networks." Proceedings of the AAAI Conference on Artificial Intelligence. Vol. 33. 2019.
[4] Peharz, Robert, et al. "Einsum Networks: Fast and Scalable Learning of Tractable Probabilistic Circuits." ICML (2020).

---

### Official Review · AnonReviewer1 · 2020-10-31

**Rating:** 3
**Confidence:** 4

**Review:**

==== Summary ====

The paper proposes a model that combines hierarchical Gaussian Mixture Models with a convolutional architecture, supporting both estimation and sampling. The model is trained end-to-end via SGD and is composed of 3 types of layers: standard convolutional and max-pooling layers and a newly proposed GMM layer. The latter operates by modeling the stack of channels at each spatial location as vectors sampled from a GMM. The outputs of the layer are the component probabilities at each location, followed by channel-wise normalization. The loss function is the average log-likelihood of every location at every GMM layer. The paper argues for using it as an alternative to other, less interpretable, probabilistic models of images and demonstrates its capacity to model the MNIST and FashionMNIST datasets.

==== Detailed Review ====

Main strengths:
* A novel architecture inspired by both ConvNets and hierarchical GMMs, allowing for more interpretable representation of images.
* Demonstrates that the model can handle simple image datasets and provides the code to reproduce the results.

Main weaknesses:
* There are no obvious theoretical advantages over other probabilistic models.
* Experiments do not compare to other methods beyond GMM, so it is hard to determine if there are any significant practical benefits. Additionally, the experiments are limited to only MNIST and FashionMNIST, which raises the question of whether this method is applicable to more complex datasets.
* The model does not represent a proper probability distribution. There is also a mismatch between the training objective and the sampling process.
* Missing references to other relevant probabilistic models of images that combine ConvNets and GMM.

I do not recommend acceptance due to the lack of theoretical or practical benefit of the proposed method and the lack of appropriate comparisons to prior approaches. In more detail:
1. The paper does not argue for any advantage to the proposed method over the alternatives beyond a general claim of interpretability. The experiments merely demonstrate that the method can model very simple image datasets and has a basic ability to detect outliers. Many models can accomplish the same, and yet they are not compared. The authors should explain why someone would prefer using this model over the alternatives (GAN, VAE, autoregressive models like PixelCNN, or even proper hierarchical graphical models).
2. The model itself is not a proper distribution, as opposed to GAN, VAE, and autoregressive models, which do represent distributions. There is a lack of theoretical justification for the proposed loss function and its connection to the generative process (I believe you might be able to show your objective is a lower bound on the true log-likelihood). Regardless, if the only measure of success is image representation, then there are other non-probabilistic methods the authors could have compared to (e.g., plain AE or Generative Latent Optimization).
3. It is claimed to be the first method to combine ConvNets and GMMs in an end-to-end manner, but prior works have already done this, though using different constructions. Specifically, Sum-Product Networks with Gaussian leaves [1,2,3,4] have been trained with convolutional architectures [2, 4] and SGD end-to-end [2, 3, 4] on several image datasets, including MNIST, FashionMNISt. These models are proper distributions, equally interpretable, and their samples are comparable to those produced in this paper. They also do not scale well to more complex image datasets (to the best of my knowledge), which is why it is so essential to show experiments on other datasets beyond these basic ones.

[1] Sum-Product Networks: A New Deep Architecture. Pool et al., 2012.
[2] Tensorial Mixture Models. Sharir et al., 2016.
[3] Deep Convolutional Sum-Product Networks. Butz et al., 2019.
[4] Random Sum-Product Networks: A Simple and Effective Approach to Probabilistic Deep Learning. Peharz et al., 2020.

---

### Decision · Program_Chairs · 2021-01-07
**Final Decision**

**Decision:**

Reject

**Comment:**

This is a clear reject. None of the reviewers supports publication of this work. The concerns of the reviewers are largely valid.